



# Brief communication: Seismological analysis of flood dynamics and hydrologically-triggered earthquake swarms associated with storm Alex

Małgorzata Chmiel[1,2,6], Maxime Godano[1], Marco Piantini[3], Pierre Brigode[1,5], Florent Gimbert[3], Maarten Bakker[3], Françoise Courboulex[1], Jean-Paul Ampuero[1], Diane Rivet[1], Anthony Sladen[1], David Ambrois[1], and Margot Chapuis[4]

[1]Université Côte d'Azur, Observatoire de la Côte d'Azur, Géoazur, CNRS, IRD, Sophia Antipolis, France
[2]Laboratory of Hydraulics, Hydrology and Glaciology, ETH Zürich, Zürich, Switzerland
[3]Institute for Geosciences and Environmental Research (IGE), CNRS / INSU, IRD, University Grenoble Alpes and Grenoble-INP, Grenoble, France
[4]Université Côte d'Azur, CNRS, ESPACE, bd Edouard Herriot, 06204 Nice, France
[5]Université Paris-Saclay, INRAE, UR HYCAR, 1 Rue Pierre-Gilles de Gennes, 92160 Antony, France
[6]Swiss Federal Institute for Forest, Snow and Landscape Research, Zürich, Switzerland

**Correspondence:** malgorzata.chmiel@wsl.ch

**Abstract.**

On October 2, 2020, the Maritime Alps in southern France were struck by the devastating storm Alex that caused locally more than 600 mm of rain in less than 24 hours. The extreme rainfall and flooding destroyed regional rain and stream gauges. That hinders our understanding of the spatial and temporal dynamics of rainfall-runoff processes during the storm. Here, we
5  show that seismological observations from permanent seismic stations constrain these processes at a catchment scale. The analysis of seismic power, peak frequency, and backazimuth provide us with the timing and velocity of the propagation of flash-flood waves associated with bedload-dominated phases of the flood on the Vésubie river. Moreover, the combined short-term average to long-term average ratio and template matching earthquake detection reveal that 114 local earthquakes between local magnitude ML=-0.5 and ML=2 were triggered by the hydrological loading and/or the resulting in-situ underground pore
10  pressure increase. This study shows the impact of storm Alex on the Earth's surface and deep layer processes and paves the way to future works that can reveal further details of these processes.

## 1  Introduction

Extreme weather events might trigger an extreme response of the Earth's surface and subsurface processes, e.g., in the form of rapid and disastrous flash-floods (e.g., Khajehei et al., 2020), mass movements (Stoffel and Huggel, 2012), and/or seismogenic
15  underground stress changes (e.g., Rigo et al., 2008). These processes contribute to societal and environmental risks and are an important agent in landscape evolution. Moreover, some extreme weather events might become more frequent due to climate change (IPCC, In Press.). That is why it is crucial to reliably quantify the spatio-temporal response of the Earth's systems to extreme weather forcing.





Seismic methods have the potential to monitor surface and subsurface processes associated with extreme weather events.
In particular, both turbulent flow and sediment transport during floods generate ground motion in different frequency bands
(Schmandt et al., 2013; Gimbert et al., 2014) that can be used to track the flood dynamics (e.g., Cook et al., 2018). Surface
seismic waves are generated by impact forces exerted by mobile particles on the river bed (e.g., Tsai et al., 2012; Gimbert
et al., 2019) and ambient seismic measurements have recently been used to monitor fluxes associated with transported bed
material Bakker et al. (2020); Lagarde et al. (2021). In the past decade, near-river seismic monitoring has been conducted
during moderate-magnitude floods (e.g., Burtin et al., 2016; Roth et al., 2016) and controlled small-magnitude flow events
(Schmandt et al., 2013, 2017). To date, extensive seismic investigations of large-magnitude flood events are rare and mostly
associated with glacier lake outburst floods (Cook et al., 2018; Maurer et al., 2020) and natural hazard cascade (Cook et al.,
2021). Yet, improved understanding of flood dynamics is crucial for early warning, risk mitigation, and modeling landscape
evolution (Raynaud et al., 2015; Borga et al., 2019).

Exceptionally intense rainfall can reactivate existing faults through changing crustal stress conditions due to additional fluid
mass load or in situ stress changes, resulting in hydrologically trigered earthquakes (e.g., Hainzl et al., 2006). Over the past
two decades, a growing number of studies has shown a correlation between meteorological events and earthquake activity in
various geological contexts (Costain and Bollinger, 2010). Several sites show the seasonal modulation of the seismicity due to
rainfall or snowmelt periods in Japan (Ueda and Kato, 2019); Nepal (Kundu et al., 2017); Taiwan (Hsu et al., 2021); Oregon,
USA (Saar and Manga, 2003); California, USA (Johnson et al., 2017; Montgomery-Brown et al., 2019); and Italy (D'Agostino
et al., 2018). Other observations display a punctual increase of seismic activity following an exceptional rainfall episode, for
example in the Swiss Alps (Roth et al., 1992), German Alps (Kraft et al., 2006), and southern France (Rigo et al., 2008).

Here, we present a set of seismological observations from 11 stations from the permanent French RESIF Network that cap-
tured the October 2020 extreme rainfall and flash flood caused by storm Alex (Carrega and Michelot, 2021) in the southwestern
Alps (the Maritime Alps), South-East France. This unique dataset not only allows us to study surface flash-flood related hazard,
but also the seismogenic subsurface response to an unusually intense rainfall which locally exceeded 600 mm in less than day.
Three rivers were strongly impacted by the flash floods: the Vésubie, the Roya, and the Tinée rivers (Figure1A). We first gain
insights onto the Vésubie river dynamics during the flash flood by analyzing seismic power, peak frequency, and dominant
backazimuthal orientation of seismic noise. These observation are compared with simple rainfall-runoff modeling (Brigode
et al., 2021a). Then, by using template matching we detect a series of impulsive signals that correspond to small earthquakes
[down to local magnitude (ML) of -0.5] in the area where rainfall rate in the Tinée river catchment area was maximum.
These preliminary analyses demonstrate that the seismological observations reported herein provide a better understanding
and quantification of the hydro-geological impact of extreme weather phenomena on the mountainous terrain and the related
fluvial hazards. The latter is important for catchment areas with few "classical" hydrological observations, as the Vésubie river
catchment presented here.

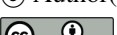



## 2 Storm Alex: a very destructive "Mediterranean episode"

On October 2, 2020, the Maritime Alps were struck by a violent meteorological event called a "Mediterranean episode" caused by storm Alex (Carrega and Michelot, 2021). Although heavy rainfalls occur regularly in autumn in the Mediterranean region, the storm Alex maximum daily rainfall was the highest measured since 1997 (Figure 1C). The rainfall started at 06:00 UTC
on October 2, 2020, lasted for less than 24 hours and generated a cumulative intensity that locally exceeded the typical yearly average (>600 mm/day, Figure 1A). These estimates have been obtained hourly with the ANTILOPE model (Laurantin, 2008) with 1 km$^2$ spatial resolution. The ANTILOPE model was produced by Météo-France and constrained by radar data and 40 rain gauges located in the region (Figure 1A), although the estimation of rainfall maps is highly uncertain in this context as a result of few rain gauges available, rainfall measurement uncertainties due to observed intensities, limits of the radar observations,
and spatial interpolation.

The torrential rains triggered hazardous flash floods and landslides of an intensity and spatial extent that have never been observed yet in this area, causing several casualties as well as large infrastructure and economic damage [Figure 1A,B1, Brigode et al. (2021b)]. To date, the spatial and temporal evolution of the flood remain poorly understood. This is partially due to a limited number of observations caused by the instrument destruction during the flood. Stream gauge measurements during
the episode are incomplete and highly uncertain due to scale saturation, destruction of measuring devices, and changes in the river bed level.

We focus our flood-dynamics analysis on the Vésubie river because (1) the Vésubie catchment has been one of the most strongly affected in the region (Figure 1, B1, B2) and (2) the seismic station coverage is particularly adequate in this catchment with three seismic stations being located in the proximity: SPIF (3-component velocimeter), BELV (3-component accelerome-
ter) and TURF (3-component velocimeter) at respectively about 1,570, 630 and 5,970 m from the Vésubie river. We investigate the level of seismic power recorded by these stations by calculating the Power Spectral Densities (PSD,Solomon (1991b)) during the storm. Then, we perform additional analysis on station SPIF by assessing temporal changes in: (1) peak frequency, (2) dominant backazimuthal orientation of seismic noise, and (3) relation between high frequency (10-45 Hz) and low frequency (1-10 Hz) seismic noise. We contextualize these seismological observations by comparing them with runoff temporal series. A
simple KLEM rainfall-runoff model (Kinematic Local Excess Model, Borga et al. (2007)) is used for runoff simulation. A full description of the methods used in this paper is provided in Appendix A.

**Figure 1.** Study area, rainfall measurements, and recorded seismic data. A. Study site and permanent RESIF seismological stations super-imposed over the total rainfall amount estimated by the ANTILOPE database (from 06:00 UTC October 2, 2020 to 06:00 UTC October 3, 2020). Background map source: © Google Maps 2021. The location of the earthquake swarms studied here is indicated by a purple star. Location in France of the study area is marked by a blue dot in the inset map. B. The village of Saint-Martin Vésubie before (summer, 2020) and after the storm episode (source: IGN 2020). C. Annual maximum daily rainfall rate from the COMEPHORE database (hourly rainfall estimated on pixels of 1 km$^2$, available since 1997, Tabary et al. (2012)) calculated over the 25 km$^2$ rectangle shown in Figure B2A. D. Vertical ground velocity at stations SPIF and TURF and vertical ground acceleration at station BELV filtered in 1-20 Hz (top) and their seismic power (bottom) recorded during 48 hours, between 00:00 October 2 and 00:00 October 4. The frequency axes are limited to 1-20 Hz.





.We contextualize our observation using a rainfall estimation obtained with the ANTILOPE model (Laurantin, 2008). The ANTILOPE model was produced by Météo-France based on radar data and 40 rain gauges located in the region (Figure 1A). The estimation of rainfall maps is highly uncertain in this context (few rain gauges available, rainfall measurement uncertainties
due to observed intensities, limits of the radar observations, and spatial interpolation).

## 3 Results

### 3.1 Seismological observations associated with the flash-flood

All three seismic stations (SPIF, BELV, and TURF) show elevated noise levels during the 24 h period starting at 07:00 UTC October 2, 2020, that overlap with the duration of storm Alex (Carrega and Michelot, 2021) (Figure 1D). The stations SPIF
and BELV show elevated seismic power (PSD) from ∼ 10:00 UTC October 2 to ∼06:00 UTC October 3 in the frequency band 1-20 Hz. The seismic power during storm Alex is 20 dB higher than the pre-flood "background" ambient seismic noise power levels. Since the decibel scale is a base-10 logarithmic scale, the 20 dB observed difference means 100 times higher seismic power. For the TURF station, the seismic power is increased by about 10 times relative to the pre-flood conditions, mostly at frequencies lower than 5 Hz. The seismic power averaged in the 1-20 Hz frequency band for SPIF and BELV stations (Figure
2A-B) show a rapid increase in recorded seismic power from 10:00 and 11:00 UTC, respectively. Three local seismic power maxima are visible on SPIF and BELV stations. Their arrival times are marked in color in Figure 2 and the seismic power thresholds used to define the maxima are shown in Figure B3.

The first two seismic power maxima have pronounced high-amplitude peaks and arrive at 12:05 and 13:15 (SPIF), and 12:30 and 13:35 UTC (BELV), respectively. The third maximum has a distributed amplitude in time and occurs between ∼16:00
and ∼20:00 UTC at SPIF and ∼16:00 and ∼22:00 UTC at BELV. The seismic power recorded at the TURF station shows a progressive increase with a single broad peak between ∼17:30 and ∼22:00 UTC. The peak associated with the first maximum has the highest magnitude at the SPIF station, while all three maxima have similar magnitudes on the BELV station. The peak associated with the first maximum lasts for ∼ 30 min and that associated with the second maximum lasts for ∼ 90 min. The peak associated with the third maximum is the broadest, lasting for 4 and 6 hours on the SPIF and BELV stations, respectively.
For the sake of comparison with the runoff modelling, we use a linear scale for the seismic power representation in Figure 2A, B3. For an alternative seismic power representation in decibels [10 log10(PSD)] the reader can refer to Figure B3.

Runoff simulations show two runoff maxima at three analysed locations (Figure 2A-C). The analysed locations correspond to the river points with the shortest distance between the seismic stations and the Vésubie river and are shown in Figure B2A. Modelling predictions indicate that the runoff maxima occur at 14:00, 14:25, and 15:00 UTC (the first runoff maximum), and
18:00, 18:25, 19:00 UTC (the second runoff maximum), from upstream to downstream. The available stream gauge measurements at Utelle (Figure B2A) show a similar rapid increase in runoff as the seismic power and the rainfall-runoff model (Figure B4). However, no maximum runoff measurements are available since the stream gauge (marked as a dark-blue diamond in Figure 1A and B2A) was destroyed during the flood.





To investigate potential changes in seismic noise sources, we calculate the peak frequency and the backazimuth (Figure

2E-F). In figure 2E peak frequency values are time color-coded meaning that each color corresponds to the consecutive 200 s long time windows shown in Figure 2A). The peak frequency corresponds to the frequency that has the maximum seismic power value in the analyzed time window (Figure B5D). Peak frequency and backazimuth ($\theta$, averaged in the 3-8 Hz frequency band, Figure 2F) show distinct value shift at the SPIF station before and during the flood. Starting from 08:30 UTC multiple lightning strikes occurred at the distance of 15 km from the SPIF station (https://www.blitzortung.org/en/, Figure B5). At

this time there are higher-amplitude arrivals visible at the SPIF station causing jumps in the peak frequency from 2 Hz to higher values up to 40-50 Hz at 09:30 UTC (Figure B5). These arrivals can be associated with the lightning/thunder, rain, or anthropogenic activity. However, at 11:00 UTC the peak frequency stabilizes at 6 Hz. Then, the peak frequency drops to 4 Hz at ~13:20 and comes back to 6 Hz at ~15:00 UTC. This drop in the peak frequency coincides in time with the second seismic power maximum visible at the SPIF station. The backazimuth starts pointing along a 100°-120° axis at 10:00 UTC (Figure 2F)

although the degree of polarization is relatively weak ($\beta^2$ ~0.5, Figure B6). The dominant degree of polarization ($\beta^2$ in the range 0-1), based on Koper and Hawley (2010), provides a measure for the confidence with which the horizontal azimuth can be interpreted, where $\beta^2$>0.5 is recommended by Goodling et al. (2018). Therefore the backazimuth observations should be taken with caution.

The relative contributions of low- (2-10 Hz) and high- (10-45 Hz) frequency seismic power are shown in Figure 2G. Different

time periods characterized by a varying relationship between low frequency and high frequency seismic power can be identified: between 08:30 and 10:00 UTC the seismic power increases similarly in the two-frequency range (slope ~1), between 10:00 and 16:00 UTC the high frequency seismic power increases more strongly (slope >1), and finally between 16:00 UTC October 2 and 07:00 UTC October 3 the seismic power decreases similarly. The equivalent of Figure 2G in the linear amplitude scale [(m/s)$^2$ Hz$^{-1}$] is presented in Figure B7. We discuss the significance of slope changes in the discussion section.

**Figure 2.** Analysis of continuous seismic signals recorded during storm Alex. Seismic power (PSD) recorded at stations A. SPIF, B. BELV, and C. TURF. The results of the runoff simulation are marked in yellow (CN60), light green (CN70), and green (CN80), where CN denotes three different basin saturation scenarios: CN70 (moderate saturation), CN60, and CN50 (rather dry conditions). Seismic power is smoothed with a moving time window of 30 min and the runoff is calculated with a 5-min time step. D. Vertical ground velocity recorded at the SPIF station filtered in 1-50 Hz. E. Peak frequency calculated for each 200 s segment. Peak frequency and corresponding time segment are marked in the same color. F. Backazimuth (smoothed over three 30-min consecutive time windows) calculated at the SPIF station averaged over 3-8 Hz and its standard deviations (dashed lines). G. Seismic power in the 2-10 Hz frequency band versus seismic power in 10-45 Hz at SPIF station. All results are shown from 07:00 UTC October 2 to 07:00 UTC October 3, 2020.

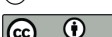



## 3.2 Earthquake swarm detection

Since 2014, the seismic activity of the studied area is permanently monitored by the Seiscomp3 (Hanka et al., 2010) system. A routine short-time average long-time average (STA/LTA) detection method is implemented in the SeisComP3 system, operating in Observatoire de la Côte d'Azur for the monitoring of the seismic activity in the South-Western Alps. For the past 7 years, the area of the Tinée valley has shown a regular monthly seismic activity with an average of 2 earthquakes of local magnitude (ML) larger than 0.4 and transient increases up to 11 earthquakes (Figure 3). However, in October 2020, 23 earthquakes were detected, which is the highest monthly earthquake rate since 2014. This uncommon seismic activity consisted of 23 earthquakes located along the Tinée valley at around 4 km depth (Figure 3B-C). The seismic crisis started on October 4, 2020, about 24 hours after the end of storm Alex, and lasted throughout October with small events in November and December (Figure 3D). The earthquakes form three distinct swarms in space and time that were mostly successively activated from South to North (Figure 3B-C). The location error is estimated to be about ±2km. We detected 91 additional earthquakes by applying the template matching detection method (Gibbons and Ringdal, 2006) to the continuous data recorded by the MVIF station (Figure 3D). The template matching increases the number of detected earthquakes by about 400 % and decreases the minimum magnitude by one unit compared to the Seiscomp3 detections based on STA/LTA. Most of the newly identified events occur on October 8 and may be related to the middle swarm since they best correlate with one of the templates constituting this cluster.

# 4 Discussion

## 4.1 The Vésubie river dynamics during the flash flood

Comparison amongst the increased seismic power (100 times larger than common noise levels), runoff modeling, and runoff measurements indicates that the signals recorded by SPIF, BELV and TURF stations during storm Alex are mostly generated by the flash flood on the Vésubie river. The rapid increase in seismic power, the changes in peak frequencies, and dominant backazimuth suggest the flash flood on the Vésubie river started at about 10:00 UTC. The backazimuth values measured at the SPIF station points towards 110° direction (Figure B2, black arrow), which does not point towards the closest river section (located at a backazimuth of 66°). The backazimuth of ∼110° may be associated with a bending of the Vésubie river channel, a ∼2.5 km long downstream reach of the Vésubie river that aligns with the estimated azimuth, or the confluence of the Venanson stream with the Vésubie river which lies in the estimated direction (Figure B2B). This provides evidence that the commonly made assumption that the recorded seismic signals are associated with the river segments located closest to the station (e.g., Zhang et al., 2021) may not be always valid.



**Figure 3.** Seismic activity of the Tinée valley. A. Monthly seismic activity between January 2014 and March 2021 detected by the SeisComP3 system. The blue arrow indicates the occurrence of storm Alex. B. Map of the seismicity over the period 2014-2021 located by SeisComP3 system. Colored circles are earthquakes following storm Alex. White circles are background seismicity. Orange triangle represents the seismological broadband station MVIF. C. North-South cross section displaying the depth range of the seismicity with respect to the sea level and the line is the average elevation of the map in B. D. Daily rate (left axis) and local magnitude (right axis) for the seismic activity following storm Alex. Grey bars are the number of earthquakes detected by the Seiscomp3 system. Orange bars are the number of earthquakes detected by template matching. Black circles represent the detected earthquakes. The size is proportional to the magnitude. Dashed blue lines indicate the duration of storm Alex.





Both seismic power and peak frequency are site-dependent seismic parameters, i.e. they depend on seismic quality factor, the velocity of Rayleigh waves, and the source-station distance (Aki and Richards, 2002). However, according to a modified Tsai et al. (2012) model for hazardous flow monitoring from Lai et al. (2018), the seismic power is strongly sensitive to particle sediment size and flow speed, while the peak frequency mostly depends on the distance from the seismic source to the receiver. Also, previous observations reported no significant shift in peak frequency with varying runoff (Schmandt et al., 2013; Burtin et al., 2016). Therefore, the observed drop in the peak frequency (down to 4 Hz) that temporally correlates with the occurrence of the second seismic power maximum at the SPIF station (Figure 2A, E) can potentially be generated by an additional, more distant river segment or by a slope failure. Indeed, the flash flood impacted the adjacent hill slopes through undercutting and destabilization of the riverbanks, leading to bank, road, and bridge collapses, and landslides distributed along the river network (Figure B1). Another possible explanation could be a tributary that becomes a dominant seismic source at this moment. However, the results of the rainfall-runoff modeling for large tributaries (Boréon and Madone de Fenestre river) do not confirm this hypothesis. Also, the backazimuth analysis does not show value changes during the second seismic power maximum. This can be due to (1) changes in the seismic source location that lie in the same general azimuthal direction, (2) the difference in time scale between backazimuth estimates made over 30 minutes versus peak frequency calculations made over 200 s windows, or, perhaps most likely, (3) the low degree of polarization of the surface waves due to spatial spread of the source or to wave scattering.

Since river flow turbulence is expected to preferentially generate ground motion at low frequencies compared to bedload transport (e.g., Burtin et al., 2011; Schmandt et al., 2013; Gimbert et al., 2014), the relationships between seismic power at low versus high frequencies can tell us whether our observations may be sensitive to bedload transport (Bakker et al., 2020). As the flood develops we observe a change in scaling between low- and high-frequency seismic power, materialized by a transition from a 0.8 to a 1.3 scaling exponent as high frequency seismic power becomes higher than -158 dB (Figure 2G). We interpret this observation as an indication that high frequency seismic power above the -158 dB threshold is mostly bedload induced. This is consistent with expectation of enhanced bedload transport from this stage onwards due to increased bed shear stress and/or the activation of additional sediment supply sources from river bed destabilization or bank erosion (Cook et al., 2018). Interestingly, after peak seismic energy has been reached, high frequency seismic power drops drastically compared to low frequency seismic power (with a scaling exponent of about 2), consistent with an abrupt decrease in sediment transport. At night, the low- versus high-frequency power scaling relation comes back to that observed during the early rising phase, consistent with higher frequencies over this timeframe getting back to being mostly sensitive to water flow. We also note that after the flood, the low-frequency seismic power is higher compared to before the flood (∼10 dB difference, see also the spectrogram of SPIF station in Figure 1), which could be due to flood-induced changes in river bed geometry and/or flow conditions (e.g. river roughness, Roth et al. (2017)) that may preferentially affect low frequency power.

About six hours passed between the beginning of storm Alex and the first flash-flood peak flow. The two seismic power maxima visible at the near-river stations (SPIF and BELV; the first maximum is marked in pink, and the second one in orange in Figures 2A, B and G) occurred in what we identified as the bedload transport phase in Figure 2G. Under the hypothesis that the two peaks associated with seismic power maxima represent the same moving source, we estimate their propagation





velocity at 5.8 (+/-1.2) m/s and 4.8 (+/-1.5) m/s, respectively. The details of the propagation velocity calculation and the error propagation calculation are given in appendix B. These peaks overlap in time with the first maximum of runoff simulations (Figure 2A-B). Such elevated and short-live peaks could be generated by flood waves. Similar peaks in seismic power generated by flood waves were observed during glacial lake outburst floods in the Himalayas by Cook et al. (2018) and Maurer et al. (2020). These peaks may share similar characteristics to sediment pulses reproduced experimentally by Piantini et al. (2021) in a torrential river setting as investigated here. Such sediment pulses can be generated by sudden destabilization of debris deposits produced by mass wasting and accumulated at the based of slopes and cliffs.

The absence of the two main maxima on the TURF station can be related to a lack of sensitivity of this station to the bedload transport due to its large distance from the river (~6 km). Farther distance means stronger geometrical attenuation at higher frequencies versus lower frequencies, and thus lower sensitivity to bedload compared to water flow (Gimbert et al., 2014). Moreover, due to the location of the TURF station further to the east, this station can be also influenced by the flood on the Roya river that is located ~10 km away from the station. The timing of the main seismic power maximum at the TURF station and the third seismic power maximum of the BELV station are well correlated with the runoff simulations and can be related to the maximum runoff. From maximum 1 to 3 there is a shift from short-lived peaks to a much more spread distribution of power through time. That could be potentially related to different dynamics of the first two maxima (associated with two fast propagating flood waves causing a sudden rise in seismic power) and a progressive increase in the seismic power associated with a progressive increase in the runoff. Finally, the differences between the observed seismic power and the runoff simulations indicate that the simple runoff simulation cannot fully explain the flash-flood dynamics.

## 4.2 Earthquake swarm in the Tinée valley

The spatial coincidence between the maximum rainfall of storm Alex in the Tinée valley and the seismic sequence a few hours later (Figure 1A) raises the question of whether the earthquakes were triggered by the heavy rainfall. Three different hypotheses can be proposed for the triggering of seismicity by meteorological forcing. The first hypothesis is a pore pressure increase at depth caused by fluid migration from the surface through hydraulically connected fractures. In this case, the time lag between rainfall at the surface and earthquakes at depth is dependent on the hydraulic diffusivity along with the fractures (Saar and Manga, 2003; Kraft et al., 2006). The second hypothesis is an elastic stress perturbation in the crust induced by hydrological loadings, such as groundwater level increase after rainfall (Rigo et al., 2008). The third hypothesis is a pore pressure increase in deep fluid-saturated poroelastic rocks in response to overlying hydrological loading (Miller, 2008; D'Agostino et al., 2018).

The time lag between the onset of the rain (October 02, 06:00 UTC) and the onset of the first earthquake swarm (October 4, 00:52 UTC, southern swarm) is $\Delta t$=43 h. Taking a seismicity depth of z=5,000m $\pm$ 2,000 m below the surface and using a time-distance dependent equation for a propagating pore pressure front, $z = \sqrt{4\pi D \Delta t}$ (Shapiro et al., 1997), we find a hydraulic diffusivity ranging from D=4.6 to 25.2 m²/s. This diffusivity range is unrealistically large to indicate earthquakes triggered by fluid migration. Indeed, with such a mechanism, earthquake activity following exceptional rainfall episodes or snowmelt is characterized by a delay of several days to several months and a lower hydraulic diffusivity ranging from D=0.01 to 5 m²/s





(Kraft et al., 2006; Saar and Manga, 2003; Husen et al., 2007; Montgomery-Brown et al., 2019). Thus, the first hypothesis is
unlikely for the first earthquake swarm.

The geology of the Tinée valley consists of limestone formations topped by a sandstone layer (Grès d'Annot). These rocks
can store large volumes of water, which might support the hypothesis of seismicity triggered by groundwater weight. Rigo
et al. (2008) describes earthquake triggering in a karstic (made up of limestone) region at depths smaller than 10 km, 43 h
after the onset of heavy rainfall. The authors interpret this earthquake activity as the response of the crust to an elastic stress
increase caused by a vertical loading because of the groundwater level rise. On the other hand, Miller (2008) shows that a sharp
increase of the hydraulic loading in karst can also produce an instantaneous increase of the pore pressure in the underlying
fluid-saturated crust, able to trigger earthquakes.

The central and northern swarms occur with a larger time delay than the first swarm, 6 and 22 days respectively (5.6 mm
and 36.0 mm of cumulative rain after storm Alex), which may be more compatible with the surface to depth fluid migration
mechanism. Yet, as these swarms are at the same depth as the southern swarm, this would imply a rather large spatial variation
of the hydraulic diffusivity from D=1.4-7.5 m$^2$/s for the middle swarm to D=0.4-2 m$^2$/s for the northern swarm. Alternatively,
the successive activation of the swarms might highlight the horizontal propagation of a pore pressure front coming from the
southern cluster area and propagating northward along a permeable pathway such as a fault parallel to the valley. The migration
velocity of the earthquake swarms from the south to the north is around 20-30 m/h, which is comparable to the values given
by Chen et al. (2012) for fluid-driven earthquakes migration. The pore pressure front could be possibly initiated by a pore
pressure increase in the hypocentral area of the southern swarm in response to the overlying hydrological loading (hypothesis
two). Therefore, we conclude that the short-term between rainfall and the southern swarm (Δt=43h) is likely compatible with
hypotheses two or three.

## 5  Conclusions

Our results show that seismometers can constrain interaction between the different Earth's systems, time- and space-dependent
processes during the flood and rainfall-runoff relationship at the catchment scale. That is particularly important in the absence
of traditional hydrological measurements, as the study presented here. Observations from permanent seismological stations
in the Maritime Alps provide the timing and velocity propagation of the flood waves. They reveal bedload and turbulence
dominated phases of the flood that occurred on the Vésubie River. Our observations also suggest that 114 earthquakes between
local magnitude ML 0.5 and 2.5 were triggered by the hydrological loading and/or by the resulting in-situ pore pressure increase
in the Tinée valley. Heavy rainfall occurs regularly in autumn in the Mediterranean region, and its intensity is increasing due
to climate change (Tramblay and Somot, 2018; Ribes et al., 2019). In the future, the installation of a seismic array dedicated to
flood could help further detect and constrain flood dynamics and triggered earthquakes (e.g., Meng and Ben-Zion, 2017; Eibl
et al., 2020; Chmiel et al., 2021). Finally, the results from this study pave the way to further analysis based on the presented
unique dataset to study surface and deep earth processes associated with storm Alex.





*Code and data availability.* Obspy Python routines (www.obspy.org) were used to download waveforms and pre-process seismic data.The seismic data is collected under the network code FR (10.15778/RESIF.FR, SPIF, TURF, MVIF stations) and RA (10.15778/RESIF.RA, BELV station) and all seismic data are openly available in the archives of French seismological and geodetic network Résif (https://seismology.resif.fr/).

The code used for backazimuth analysis can be found in the online supplement of Goodling et al. (2018) paper. Rainfall data (ANTILOPE and COMEPHORE) were provided by Météo-France and are available on request. To gain access please contact Pierre Brigode.

## Appendix A: Methods

### A1  Seismic power calculation and peak frequency

Stations SPIF, BELV, and TURF are located 1,570, 630, and 5,970 m away from the Vésubie river, respectively. SPIF and

TURF are equipped with a 3-component broadband (BB) velocimeter and a 3-component accelerometer, and BELV is only equipped with a 3-component accelerometer. On SPIF and TURF we use the BB recordings for the analysis because of its higher sensitivity than the accelerometer. The sampling frequency for SPIF and TURF stations is 100 Hz, and BELV station 125 Hz. Stations BELV and TURF are affected by high-frequency noise that does not allow us to analyse signals higher than 20 Hz. To focus on river-generated seismic signals, we use high-frequency signals (1-50 Hz for SPIF station, and 1-20 Hz

for BELV and TURF stations). After the removal of the instrumental response, we first calculate Power Spectral Densities (PSD) splitting the data into 200 s-long time windows with a 50% overlap (in Figures 1D and 2A-C), but no overlap is used in Figures 2E and 2G. The windowing function window is applied to each segment and the PSD is calculated by Welch's average periodogram method (Solomon, 1991a). Then for the SPIF station, we follow previous work on debris flows (Lai et al., 2018) and we investigate signal's peak frequency in individual 200 s time windows between 2-50 Hz. We also analyse seismic power

recorded on the SPIF station in two different frequencies bands: 2-10 Hz and 10-45 Hz. For that, we estimate PSD using again Welch's method with time segments of 2 s and no overlap, and then we calculate a median over 30 s time windows.

### A2  Azimuth analysis

We perform a frequency-dependent polarization analysis to determine the dominant backazimuth assuming that the seismic signature of the flood is dominated by surface waves on the SPIF station (Goodling et al., 2018). The horizontal azimuth and

degree of polarization are determined based on the dominant eigenvector of the spectral covariance matrix of the 3 measured components (N, E and Z), following the approach of Park et al. (2005) and its recent application by Goodling et al. (2018). We determine these variables for 30-minute intervals using 9 subwindows with 50% overlap. The dominant azimuth per frequency ($\theta$) is obtained and given for a range 0-180° as there is a 180° ambiguity in this value.

### A3  Rainfall-runoff models

Runoff is firstly estimated using the Soil Conservation Service-Curve Number (SCS-CN) production function method. The SCS-CN function allows to estimate the runoff from a rainfall event depending on the catchment saturation conditions. A simplified unit hydrograph routing function is then used to produce temporal runoff series. This analysis aims at estimating,





for each studied catchment, the distances between each Digital Elevation Model (DEM) grid cell and the considered outlet and use the distance to root the runoff at the studied catchment outlets. A distinction is made between the distance travelled on the slopes and the distance travelled in the river (i.e., within the hydrographic network): the flow velocity on the slopes (fixed here at 0.2 m/s) is assumed to be slower than that in the river (fixed here at 5 m/s). These distances are used to calculate, for each grid cell "x" belonging to a studied watershed, the transfer time $\tau$ [in (s)] between this grid cell x and the considered outlet:

$$\tau(x) = \frac{L_h(x)}{v_h} + \frac{L_c(x)}{v_c} \tag{A1}$$

where: $L_h(x)$: distance (on the slopes) between the grid cell "x" and the considered catchment outlet (m), $v_h$: flow velocity on the slopes (m/s), $L_c(x)$: distance (in the river network) between the grid cell "x" and the considered catchment outlet [m], $v_c$: flow velocity in the river network (m/s).

These transfer times are used to calculate the simulated flow, at time step t, at each studied outlet (denoted Q and expressed in m$^3$/s) by the following expression (no initial base flow is considered in this study):

$$Q(t) = \int_A q(t - \tau(x), x) dx \tag{A2}$$

where: $A$: catchment area upstream of the grid cell "x" (km$^2$), $q$: runoff estimated at timestep "t" and at the grid cell "x" (m/s). The runoff is simulated for three locations along the Vésubie river which are the closest to the seismic stations (Figure B2).

## A4 Earthquake swarm detection

Previous studies have shown that template matching (e.g., Gibbons and Ringdal, 2006) has a higher detection sensitivity than threshold-based methods such as the STA/LTA used in the Seiscomp3 system. We use template matching to detect low-magnitude earthquakes that belong to the earthquake swarms. Template matching is performed on the broadband station MVIF (10 km to the south of the swarm). We verified that this station was little affected by the seismic noise generated by the increased river flow during and after the storm. We use the following approach. Data are bandpass filtered in the 5-30 Hz frequency band. We use as templates the 23 earthquakes detected by SeisComP3. The templates are constructed using a 5-s window that includes P and S waves. Next, each template is cross-correlated with daily continuous seismic data. We use only vertical components of the seismograms and we automatically scan the seismic data between September 27 and December 10, 2020. A new earthquake is detected if the cross-correlation coefficient exceeds a threshold of 0.6. This value allows the detection of earthquake waveforms that might slightly differ from the templates (if, for example, the origin location is not the same) while minimizing the number of false detections. Finally, the magnitude of detected earthquakes is estimated from the ratio between its maximum amplitude and the maximum amplitude of the best-correlated template (local magnitude ML). An example of templates and detected events by template matching are presented in Figure B9-B11.





## A5 Peak propagation velocity and uncertainty calculation

The peak arrival times are manually picked by taking the beginning of the maximum above fixed seismic power (PSD) thresholds (Figure B3,B8). Also, we verify the time delay between the two PSDs using cross-correlation (Figure B8). We find two

maxima of 0.3 and 0.15 at time lag values of 19 and 28 min, respectively. We calculate the peak propagation velocity as a ratio between the distance (d) of the two nearest river coordinates to the SPIF and BELV station (8,012 m) to manually pick the propagation time of the peaks (t). To calculate the distance, we use the nearest river coordinates to the stations, and we integrate the distance following the Vésubie river coordinates (8,012 m). Then, we use error propagation to estimate the uncertainty of the estimated velocity propagation. For that, we use the variance formula assuming that the distance and time measurements

are independent:

$$s_v = \sqrt{\left(-\frac{d}{t^2}\right)^2 s_t^2 + \left(\frac{1}{t}\right)^2 s_d^2} \tag{A3}$$

where $d$ is the distance between the two nearest river coordinates to the SPIF and BELV station (8,012 m), $t$ the manually picked propagation time of the peaks (s), $s_t$ the standard variation calculated of the three propagation times (s): (1) the manually picked propagation time of the peaks, and (2) the two cross-correlation calculated propagation times,, $s_d$ the standard deviation of the

two distances (m): (1) the distance between the nearest river coordinates to the SPIF and the BELV stations (8,012 m), and (2) the distance of the closest river segment that aligns with the dominant backazimuth calculated at the SPIF station to the closest river coordinates to the BELV station (5,512 m)).



## Appendix B: Supplemental figures

**Figure B1.** The consequences of storm Alex in the Maritime Alps. A. Landslide located upstream from the village of Saint Martin-Vésubie on the right bank of the Boréon river. B. Bank collapse next to the village of Saint Martin-Vésubie. C. Aerial view on the village of Roquebillière Vieux. D. Partial bank collapse and deposited material next to Louvivier village. Photo credits: Florent Adamo/Cerema.


**Figure B2.** A. Map section showing the Vésubie and the Boréon river. The 25 km² square used for the rainfall calculation in Figure 1B is shown with black dashed lines. Background map source: © Google Maps 2021. B. Zoom on the square marked in panel A. Three dominant azimuth are indicated in yellow arrows showing dominant noise directions of 100°, 110°, and 120° degrees (source: IGN 2020). C. Zoom on the intersection between the Venanson stream and the Vésubie river, with a slope failure indicated that is adjacent to the Venanson stream (source: IGN 2020).





**Figure B3.** Seismic power (PSD) recorded at SPIF, BELV, and TURF seismic stations in linear scale (left panels) and logarithmic scale (dB, right panels). The seismic power is averaged in 1-20 Hz frequency band, between 07:00 UTC October 2 and 07:00 October 3. Vertical lines show the starting hours of the three peaks, and the horizontal lines show the threshold used to define the peaks.





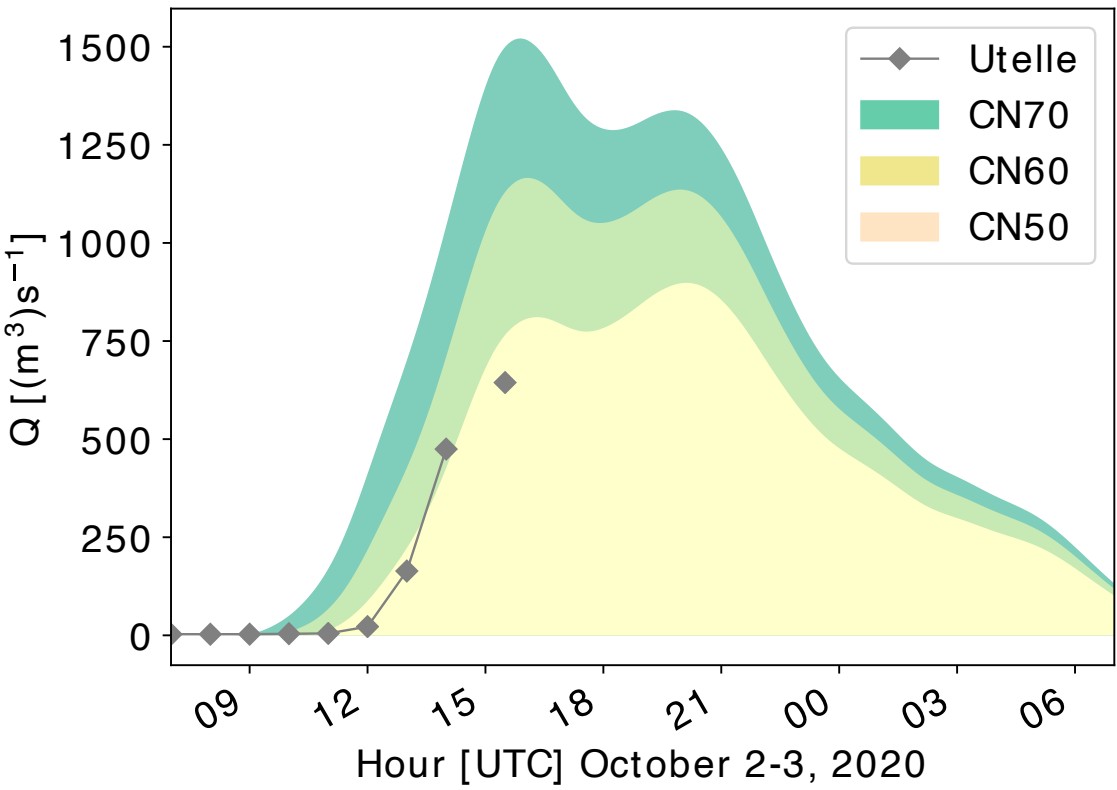

**Figure B4.** Runoff modeling for three different basin saturation scenarios: CN70 (moderate saturation), CN60, and CN50 (rather dry conditions). Available runoff measurements from the stream gauge at Utelle are presented in gray diamonds. The comparison between the stream gauge measurements and runoff modeling indicates rather dry basin conditions (CN50 scenario). However, there is an uncertainty in the runoff modeling related to the estimated flow velocities on the slope (0.2 m/s) and in the river (5.0 m/s). Moreover, the estimated runoff values are too low comparing to the damage that occurred in the Vésubie catchment.



**Figure B5.** A. Vertical ground velocity recorded filtered in 1-50 Hz. B. Peak frequency calculated for each 200 s segment. C. Rainfall measured by the rain gauge at located at Saint-Martin Vésubie. The measurement stopped when the instrument was destroyed. Lightning in the distance <15 km from the SPIF station. Each circle represent a lightening strike, the larger and the darker the circle the closer the lightening. D. Seismic power calculated in windows of 200s. Peak frequency, corresponding time segment, and seismic power (PSD) are marked in the same color.




**Figure B6.** A. Non-smoothed polarization degree $\beta^2$. The impulsive high value polarization levels at 06:00 UTC October 3 can be associated with anthropogenic noise sources and there are also visible at different days. B. Non-smoothed backazimuth direction $\theta$, averaged over 3-8 Hz. The mean is shown in continuous line, and the standard deviation in the dashed lines.

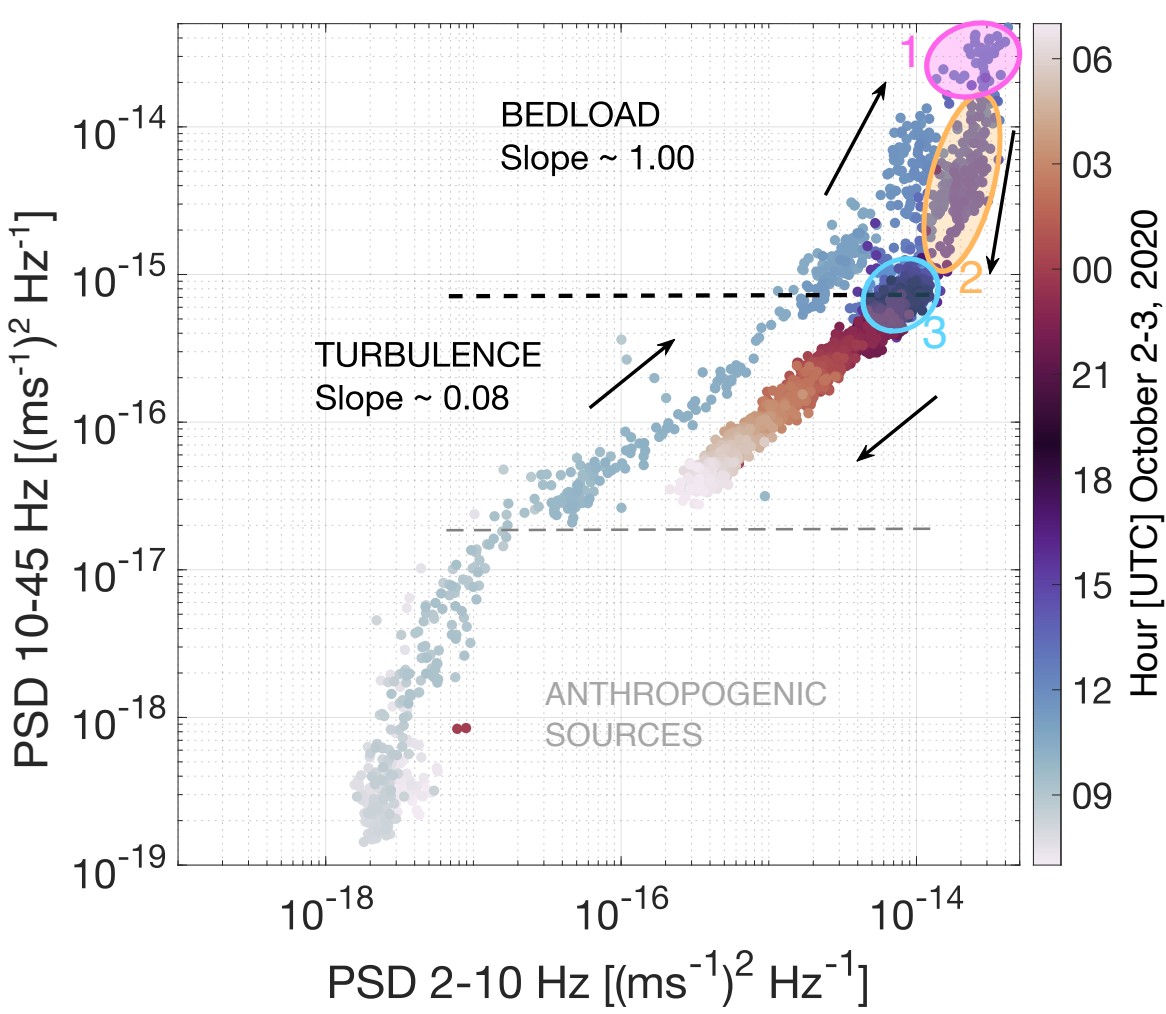

**Figure B7.** The same plot as Figure 2G, but in the linear scale. Seismic power calculated in 2-10 Hz versus seismic power calculated 10-45 Hz at the SPIF station. The seismic power peaks are marked in different colors.



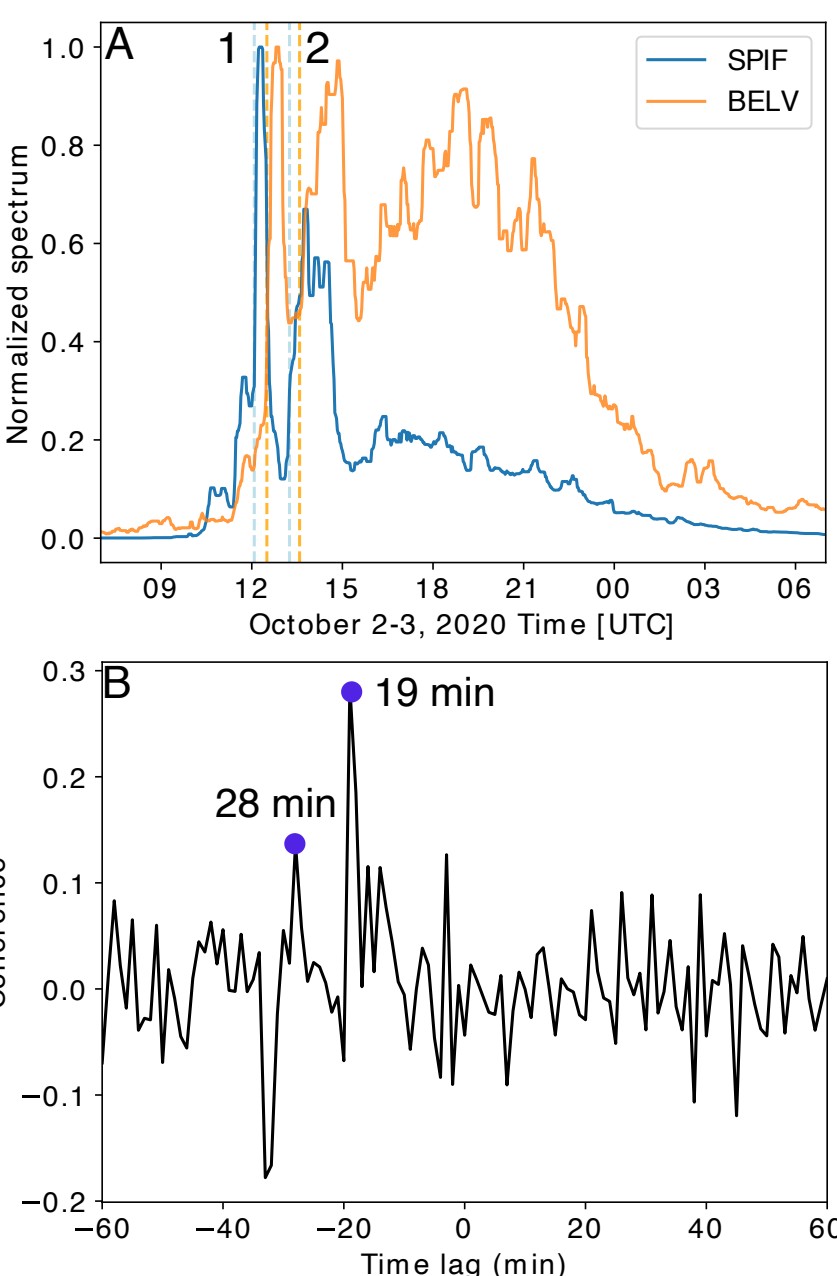

**Figure B8.** A. Normalized seismic power recorded at the SPIF and the BELV stations smoothed over 30 min moving time average. Arrivals times of peaks 1 and 2 are marked in dashed lines. B. Normalized cross-correlation (coherence) between the normalized seismic power shown in A.




**Figure B9.** Red: template located in the southern swarm and black: examples of detected events by template matching. Seismograms show vertical ground velocity recorded at the station MVIF and are bandpass-filtered in the range 5-20 Hz.




**Figure B10.** Same as Figure B9, but for a template located in the middle swarm.



**Figure B11.** Same as Figure B9, but for a template located in the northern swarm.

*Author contributions.* MC, MP, MB processed and analyzed the flood seismic data. MG performed the earthquake swarm analysis. PB

performed the rainfall-runoff modeling. FG, PB, FC, J-PA, DA, AS, DA, MC helped with the data analysis and interpretation. MC and MG

prepared the manuscript with a help of all the co-authors.

*Competing interests.* The authors declare that they have no conflict of interest.



*Acknowledgements.* We thank Observatoire de la Côte d'Azur that partially founded this work. We also thank the ANR project ANR SEIS-MORIV ANR-17-CE01-0008 granted to FG. We thank Didier Brunel, Christophe Maron, Jerome Cheze and all the persons in charge of the permanent seismic network and data. We thank Florent Adamo and Cerema for kindly allowing us to use their photos.






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
