# Peer review of "Brief communication: Seismological analysis of flood dynamics and hydrologically-triggered earthquake swarms associated with Storm Alex"

_Natural Hazards and Earth System Sciences, 2021_

## Referee Comment (RC1)

**Review of the manuscript "Seismological analysis of flood dynamics and hydrologically-triggered earthquake swarms associated with storm Alex " by Chmiel et al. for publication in NHESS.**

**General comments**

The article presents and analyses seismological records at permanent stations related to the storm Alex, which hit the southeastern France in October 2020. This storm was particularly damaging with important flash-flood waves in the rivers, in particular the Vésubie river. The seismological analyses consist of the temporal evolution of the seismic power, the peak frequency and the backazimuths owing to follow the flood propagation. The recordings alos make it possible to detect induced seismicity as three swarms triggered several days after the storm.

This paper is of great scientific interest, very well-written and well-illustrated. It uses seismological records in an original way in order to constrain the timing and the propagation of river floods in the area affected by this exceptional weather event. In addition, the detection of induced earthquakes highlights the importance of the rainfall in the seismic activity.
The manuscript is acceptable for publication with minor revisions as suggested hereafter.

**Specific comments**

- A threshold at each considered seismic station is used to define the seismic power maxima. It is not clear, and it is not explained, how these thresholds are determined/fixed: from the background noise? From an average of the seismic power over a time window? Please, clarify that point.
- Again about the seismic power peaks, is the third one specifically defined with the TURF station because it is not very clear on the other two stations?
- Regarding the migration velocity of the earthquakes, are you sure that the velocity is in m/h and not in m/day (line 240)? Commonly, this velocity is considered to be from 10m/d to 100m/d, and Chen et al. (2012) found a velocity at 38m/d. The velocity you determined here, implies a velocity at 500-800 m/d, which is near aseismic slip (according to Chen et al., 2012). Are you considering the migration velocity of the earthquakes inside a swarm, or the migration velocity from one swarm to another? In this last case, I am not absolutely convinced that we can relate the time occurrence of the swarms to a fluid-driven migration. For example, in the central swarm, there are events at around 10 days as in the southern swarm, but also the latest events at more than 60 days (Figure 3).
- Any comment on the fact that the "induced" seismicity by the storm is nearly at the same location of the previous background seismicity, especially at depth?
- Finally, do you think that such seismic observations can be used to improve the rainfall-runoff simulations?

**Technical corrections**

- line 24: put the citations into (…)
- line 44: these observation**s**
- Figure 1A: there is a small typing error in the color scale where 30 is for 300
- line 77: the sentence begins with a "."
- line 78: concerning the "40 rain gauges": Only one is visible in Figure 1A, and then at what distances are they?
- Please indicate more precisely that Figures B5 and B6 are for SPIF station.

---

## Author Response (AR1)

**Manuscript title:** Seismological analysis of flood dynamics and hydrologically-triggered earthquake swarms associated with Storm Alex

Manuscript ID: NHESS-2021-339

Dear Editor and dear Reviewers,

We thank two anonymous Reviewers for their helpful reviews and valuable comments. We addressed all of the comments and modified the manuscript accordingly. In the following, the Reviewer's comments are marked in blue, our responses in black, and parts of the manuscript are marked in *italic*.

Best,
Małgorzata Chmiel

**Reviewer 1**

Review of the manuscript "Seismological analysis of flood dynamics and hydrologically-triggered earthquake swarms associated with storm Alex " by Chmiel et al. for publication in NHESS.

**General comments**

The article presents and analyses seismological records at permanent stations related to the storm Alex, which hit the southeastern France in October 2020. This storm was particularly damaging with important flash-flood waves in the rivers, in particular the Vésubie river. The seismological analyses consist of the temporal evolution of the seismic power, the peak frequency and the backazimuths owing to follow the flood propagation. The recordings alos make it possible to detect induced seismicity as three swarms triggered several days after the storm. This paper is of great scientific interest, very well-written and well-illustrated. It uses seismological records in an original way in order to constrain the timing and the propagation of river floods in the area affected by this exceptional weather event. In addition, the detection of induced earthquakes highlights the importance of the rainfall in the seismic activity. The manuscript is acceptable for publication with minor revisions as suggested hereafter.

**Our response:** We thank the Reviewer for this detailed review, helpful comments, and nice words. Please find below our replies to the Reviewer's comments.

**Specific comments**

- A threshold at each considered seismic station is used to define the seismic power maxima. It is not clear, and it is not explained, how these thresholds are determined/fixed: from the background noise? From an average of the seismic power over a time window? Please, clarify that point.

**Our response:** We determine the thresholds manually; they delimit the values in seismic power when the seismic power strongly and rapidly increases. We will add the following sentence (in red) to the manuscript (Lines 91-92):

*"Their start and end times are marked in color in Figure 2, and the seismic power thresholds used to define the maxima are shown in Figure B3. We determine the thresholds manually; they delimit the values in seismic power when the seismic power strongly and rapidly increases."*

*- Again about the seismic power peaks, is the third one specifically defined with the TURF station because it is not very clear on the other two stations?*

**Our response:** In Figure 2C we only mark the third maximum of the seismic power recorded at the TURF station because we cannot identify maxima 1 and 2 at the TURF station. However, we also define the third maximum for the SPIF and the BELV station, and we describe it in Lines 95-100, and Lines 200-205. We will add the following sentence to the manuscript (Lines 91-92):

*"The maxima 1 and 2 are not marked in Figure 2C because we cannot identify them at the TURF station."*

*- Regarding the migration velocity of the earthquakes, are you sure that the velocity is in m/h and not in m/day (line 240)? Commonly, this velocity is considered to be from 10m/d to 100m/d, and Chen et al. (2012) found a velocity at 38m/d. The velocity you determined here, implies a velocity at 500-800 m/d, which is near aseismic slip (according to Chen et al., 2012). Are you considering the migration velocity of the earthquakes inside a swarm, or the migration velocity from one swarm to another? In this last case, I am not absolutely convinced that we can relate the time occurrence of the swarms to a fluid-driven migration. For example, in the central swarm, there are events at around 10 days as in the southern swarm, but also the latest events at more than 60 days (Figure 3).*

**Our response:** Thank you for this comment. Regarding the discussion on swarm migration (Lines 237-244):

We consider the migration velocity from southern swarm to central and northern swarms. The locations are not accurate enough to correctly resolve earthquake migration inside swarms.

In the first version of the manuscript, there was confusion with the units of the migration velocity, which led to an erroneous interpretation. The velocity we estimate (line 240) is well expressed in m/h. The Reviewer is right; our values of 20-30 m/h are much larger than the migration velocity usually reported for fluid diffusion, which is of the order of tens of m/day (e.g., ~38 m/day Chen et al. 2012, ~ 30 m/day Ruhl et al. 2016). On the other hand, our migration velocity is somewhat lower than values usually reported for aseismic slip-driven seismicity, typical of order 100-1000 m/h (e.g., Lohman and McGuire, 2007; Roland and McGuire, 2009; Ruhl et al., 2016; Hatch et al., 2020). However, Chen and Shearer, 2011 and Chen et al., 2012, also attribute migration speeds of orders 10-100 m/h to aseismic slip, which are values compatible with our velocities.

Hence, we rewrite Lines 237-244 and propose an alternative model explaining the northward migration:

*"The successive activation of the three swarms could also suggest an alternative mechanism of triggered seismicity. The northward migration of the seismicity is around 20-30 m/h. This velocity is much greater than 1-10 m/day, usually attributed to fluid diffusion-driven seismicity (e.g., Chen et al. 2012, Ruhl et al., 2016). Yet, this velocity is also lower than velocities reported for aseismic slip-driven seismicity (typically 100-1000 m/h, e.g., Lohman and McGuire, 2007; Roland and McGuire, 2009; Ruhl et al., 2016; Hatch et al., 2020). However, Chen and Shearer, 2011 and Chen et al., 2012, also attribute slow earthquake migration of orders 10-100 m/h to aseismic slip, which are values compatible with velocities we found. Hence the northward migration of the seismicity might highlight the horizontal propagation of an aseismic slip along a fault parallel to the valley. The overlying hydrological loading could trigger this aseismic slip, and its northward propagation could drive the successive rupture of seismic asperities corresponding to the three swarms. The interplay between hydromechanical and aseismic slip processes is increasingly recognized as a driver of earthquake swarms (e.g., De Barros et al., 2020; Hatch et al., 2020)."*

Regarding the discussion on fluid diffusion process (Lines 234-237):

We included the southern swarm in the discussion about the mechanism of fluid diffusion from surface to depth, since its resumption occurs at the same time as the activation of the central swarm.

*" The resumption of activity of the southern swarm at the same time as the activation of the central swarm (6 days after Storm Alex) and the activation of the northern swarm (22 days after Storm Alex)*

*are more compatible with surface to depth fluid migration. However, as these swarms are at the same depth, this would imply a rather large spatial variation of the hydraulic diffusivity from D=1.4-7.5 m2/s for the southern and central swarms to D=0.4-2m2/s for the northern swarm."*

- Any comment on the fact that the "induced" seismicity by the storm is nearly at the same location of the previous background seismicity, especially at depth?

**Our response:** Indeed, the seismicity following Storm Alex is at the same location as the background seismicity. As shown in Figure 3, this part of the South-western Alps experiences a regular moderate seismic activity. So, we think Storm Alex has promoted ruptures on seismogenic structures that could have failed in the longer term. Hence the storm just played the role of a trigger.

We will add a sentence explaining that (end of section 4.2):

*"Finally, the seismicity triggered by Storm Alex is collocated with the previous background seismicity, especially at depth (Figure 3). This area of the south-western Alps experiences regular moderate seismic activity. Therefore, the heavy rainfall has likely promoted ruptures on seismogenic structures that could have failed in the longer term."*

- Finally, do you think that such seismic observations can be used to improve the rainfall-runoff simulations?

**Our response:** Seismic observations can provide the timing and propagation velocity of flood peaks and estimates of the flood's start and end times. This information can provide additional constraints for more accurate rainfall-runoff simulations needed to investigate spatio-temporal flood dynamics further. For example, in this study, we adjusted the flow velocity in the runoff simulation to 5 m/s, which corresponds to the peak velocities retrieved from the seismic observations.

We will add the following sentence into the manuscript at the end of section 4.1:

*"In future works, seismic observations can provide additional constraints for more accurate rainfall-runoff simulations needed to further investigate the spatio-temporal dynamics of flash-floods."*

Technical corrections

- line 24: put the citations into (…)

**Our response:** Thank you for pointing this out, we corrected it.

- line 44: these observations

**Our response:** We corrected it.

- Figure 1A: there is a small typing error in the color scale where 30 is for 300

**Our response:** Thank you for noticing this error, we corrected the color scale.

- line 77: the sentence begins with a "."

**Our response:** We corrected it.

- line 78: concerning the "40 rain gauges": Only one is visible in Figure 1A, and then at what distances are they?

**Our response:** We added a figure in the Appendix (see below) that shows the position of the regional stream and rain gauges. The rain gauges are located at distances from 2 to 51 km from the SPIF station. The closest rain gauge to the SPIF station is shown in Figure B2. We will add the following sentence to the manuscript in Lines 57-60:

*"The ANTILOPE rainfall estimation was produced by Météo-France and constrained by radar data and 40 rain gauges located in the region (Figure 1A). The location of regional rain and stream gauges is shown in Figure B1. The estimation of rainfall maps is highly uncertain in this context due to few rain gauges available, rainfall measurement uncertainties due to observed intensities, limits of the radar observations, and spatial interpolation."*

We will also add the following sentence to the caption in Figure B5:

*"This is the closest rain gauge to the SPIF station located at the distance of 1.9 km."*

[Figure]

*Figure B1: The location of rain and stream gauges and seismic stations in southeastern France.*

- Please indicate more precisely that Figures B5 and B6 are for SPIF station.

**Our response:** We will add "SPIF" to panels A, B, and D in Figure B5 and to panels A and B in Figure B6. We will also add "SPIF" to panels A, B, and D in Figure B5. We will also add *"Analysis of seismic data recorded at the SPIF station and meteorological data"* to the caption of Figure B5 and *"Backazimuth analysis at the station SPIF"* to the caption of Figure B6.

**References:**

Chen, X., and Shearer, P. M. (2011), Comprehensive analysis of earthquake source spectra and swarms in the Salton Trough, California, J. Geophys. Res., 116, B09309, doi:10.1029/2011JB008263.

De Barros, L., Cappa, F., Deschamps, A., & Dublanchet, P. (2020). Imbricated Aseismic Slip and Fluid Diffusion Drive a Seismic Swarm in the Corinth Gulf, Greece. Geophysical Research Letters, 47, https://doi.org/10.1029/2020GL087142

Hatch, R. L., Abercrombie, R. E., Ruhl, C. J., & Smith, K. D. (2020). Evidence of aseismic and fluid-driven processes in a small complex seismic swarm near Virginia City, Nevada. Geophysical Research Letters, 47, e2019GL085477, https://doi.org/10.1029/2019GL085477

Lohman, R. B., and McGuire, J. J. (2007), Earthquake swarms driven by aseismic creep in the Salton Trough, California, J. Geophys. Res., 112, B04405, https://doi.org/10.1029/2006JB004596.

Roland, E., and McGuire, J. J. (2009), Earthquake swarms on transform faults, Geophysical Journal International, Volume 178, Issue 3, Pages 1677–1690, https://doi.org/10.1111/j.1365-246X.2009.04214.x

Ruhl, C. J., Abercrombie, R. E., Smith, K. D., and Zaliapin, I. (2016), Complex spatiotemporal evolution of the 2008 Mw 4.9 Mogul earthquake swarm (Reno, Nevada): Interplay of fluid and faulting, J. Geophys. Res. Solid Earth, 121, 8196– 8216, https://doi.org/10.1002/2016JB01339, .

**Reviewer 2**

**RC2**: 'Comment on nhess-2021-339', Anonymous Referee #2, 16 Jan 2022

First, I apologize about the long delay in returning this review.

In this paper, the authors use data from a permanent seismic network to explore the seismic signals generated by an extreme precipitation event in the south of France. The data provide insights into the flood generated on one of the highly affected rivers and identify a swarm of small earthquakes apparently triggered by the storm. I found this an interesting and nice-presented paper. Destructive floods like the one described here are indeed difficult to observe using traditional methods, and this paper demonstrates the potential of seismic methods for flood observation. As well, it's not so often that major flash floods occur within such a conveniently laid out seismic network, so it's great to see this opportunity exploited. Overall, I found this an enjoyable paper to read, and I have no major concerns to raise, just some relatively minor comments about clarifications and presentation.

**Our response:** We thank the Reviewer for the helpful review and nice words. Please find below our replies to the Reviewer's comments.

How were the thresholds for the peaks selected? It would be nice to indicate both the start and end of each peak in figure 2 – you've selected a set of points for each peak in fig. 2G, but we can't tell exactly what those correspond to in the time series. It could also be interesting to identify on the waveform and power plots where the breaks in slope in fig. 2G are. Especially with the different units, it's a little difficult to compare to panel A. Otherwise, I really like figure 2!

**Our response:** Thank you for this comment. We determine the thresholds manually; they delimit the values in seismic power when the seismic power strongly and rapidly increases. We will add the following sentence (in red) to the manuscript (Lines 91-92):

*"Their start and end times are marked in color in Figure 2, and the seismic power thresholds used to define the maxima are shown in Figure B3. We determine the thresholds manually; they delimit the values in seismic power when the seismic power strongly and rapidly increases."*

We thank the Reviewer for this helpful suggestion. We added the peak end times as vertical dashed lines to Figure 2 and Figure B5:

[Figure]

*Figure 2. Analysis of continuous seismic signals recorded during storm Alex. Seismic power (PSD) averaged between 1 and 20 Hz and recorded at stations A. SPIF, B. BELV, and C. TURF. The results of the runoff simulation are marked in yellow (CN60), light green (CN70), and green (CN80), where CN denotes three different basin saturation scenarios: CN70 (moderate saturation), CN60, and CN50 (rather dry conditions). Seismic power is smoothed with a moving time window of 30 min and the runoff is calculated with a 5-min time step. D. Vertical ground velocity recorded at the SPIF station filtered in 1-50 Hz. E. Peak frequency calculated for each 200 s segment. Peak frequency and corresponding time segment are marked in the same color. F. Backazimuth (smoothed over three 30-min consecutive time windows) calculated at the SPIF station averaged over 3-8 Hz and its standard deviations (dashed lines). G. Seismic power in the 2-10 Hz frequency band versus seismic power in 10-45 Hz at SPIF station. All results are shown from 07:00 UTC October 2 to 07:00 UTC October 3, 2020.*

[Figure]

*Figure B5. Analysis of seismic data recorded at the SPIF station and meteorological data. A. Vertical ground velocity recorded filtered in 1-50 Hz. B. Peak frequency calculated for each 200 s segment. C. Rainfall measured by the rain gauge at located at Saint-Martin Vésubie. This is the closest rain gauge to the SPIF station located at the distance of 1.9 km. The measurement stopped when the instrument was destroyed. Lightning in the distance <15 km from the SPIF station. Each circle represents a lightning strike, the larger and the darker the circle the closer the lightening. D. Seismic power calculated in windows of 200s. Peak frequency, corresponding time segment, and seismic power (PSD) are marked in the same color.*

The detection of an earthquake swarm related to the storm is quite interesting. This is not my expertise, so I can't properly comment on this aspect, but I found the discussion clear and reasonable. I do wonder if you've tried calculating dv/v over this time period? Maybe this would provide some insight into the state of the subsurface in the months after the storm?

**Our response:** Thank you for this suggestion. We agree with the Reviewer that measurements of relative seismic velocity changes (dv/v) could provide additional insights into the state of the subsurface before, during, and after the storm. Such analysis is beyond the scope of this short communication but can be performed in future studies.

We will add one sentence on this topic in the Discussion section (end of section 4.2):

*"In future studies, measurements of relative seismic velocity changes (dv/v, e.g., Brenguier et al., 2008; Illien et al., 2022) could provide additional insights into the state of the subsurface before, during, and after the storm."*

Line 54: this statement makes it sound like there was higher rainfall in 1997, but I guess 1997 was just when the record starts. This should be made clear, as it's a big difference.

**Our response:** Thank you for pointing this out. We will change this phrase to:

*"Although heavy rainfalls occur regularly in autumn in the Mediterranean region, the Storm Alex maximum daily rainfall was the highest since the beginning of the rainfall measurements in 1997. The continuous regional rainfall COMEPHORE database used in this study started in 1997, Figure 1C."*

Line 68: "particularly adequate" is a strange combination. Particularly suitable?

**Our response:** We will change this phrase to particularly suitable.

Lines 79-80: this is basically a repeat of lines 58-60

**Our response:** Thank you for noticing it, we will remove the repetition from Lines 79-80.

Line 156: Roth et al., 2016 could be cited here

**Our response:** We will add the citation of Roth et al., 2016 to this sentence.

Line 161: it's not clear how are proposing that a slope failure would cause a sustained change in the peak frequency? I could imagine a slope failure that increases the local river noise (due to sediment input, geometry changes, roughness, etc) in a more distant segment, but I'd think that the failures themselves would produce much more punctuated signals.

**Our response:** We agree with the Reviewer that a slope failure would cause punctuated signals.

We modified this part of the manuscript as follows:

*"Therefore, the observed drop in the peak frequency (down to 4 Hz) that temporally correlates with the occurrence of the second seismic power maximum at the SPIF station (Figure 2A, E) can potentially be generated by a stronger more distant source."*

Line 183: you haven't been mentioning times throughout this paragraph, so specifying night here sounds a bit funny – like there's something diurnal that makes the time of day matter, which I guess is not the intention.

**Our response:** We will change the phrase from "*At night,*" to "*Next,*".

Lines 199-205: Could the lack of peaks also be influenced by the larger sampling area of the TURF station? Because of its farther distance, it will have a similar sensitivity to a pretty long stretch of channel, which should smooth out moving peaks.

**Our response:** Thank you for this pertinent comment. We agree with the Reviewer that the lack of peaks can also be related to the sensitivity of the TURF station that samples a more extended river segment.

We will add the following sentences in Lines 199-205:

*"The absence of the two main maxima on the TURF station can be related to a lack of sensitivity of this station to the bedload transport due to its large distance from the river (~6 km). Farther distance means stronger geometrical attenuation at higher frequencies versus lower frequencies, and thus lower*

*sensitivity to bedload compared to water flow (Gimbert et al., 2014). Also, this station samples a longer river segment because of its farther distance, which could smooth out moving peaks."*

Line 237: stick with "central", not "middle"

**Our response:** Thank you, we will use "central" in this paragraph.

Fig. 1: the precipitation colors on top of the Google earth colors make this a bit busy. It is ok – we can see the precip pattern, but it might look much nicer with a hillshade for the background. In panel D, the labels on the waveform y-axes are too crowded. Maybe you could make panel C shorter to give panel D a little more space?

**Our response:** Thank you for this comment. We shifted panels C and D in Figure 1 to the right, and we also shifted panel C to the top of the figure to give panel D a bit more space.

Fig. 2: It would be helpful to have the frequency range used for A-C in the caption

**Our response:** We added the frequency range to the caption (see above).

**References:**

Brenguier F., Campillo, M., Hadziioannou, C., Shapiro, N. M., Nadeau, R. M., Larose, E. (2008), Postseismic relaxation along the San Andreas fault at Parkfield from continuous seismological observations. Science, 321(5895):1478-81, https://doi.org/10.1126/science.1160943

Illien, L., Sens-Schönfelder, C., Andermann, C., Marc, O., Cook, K. L., Adhikari, L. B., & Hovius, N. (2022). Seismic velocity recovery in the subsurface: Transient damage and groundwater drainage following the 2015 Gorkha earthquake, Nepal. Journal of Geophysical Research: Solid Earth, 127, e2021JB023402. https://doi.org/10.1029/2021JB023402